# Association between Arousals during Sleep and Subclinical Coronary Atherosclerosis in Patients with Obstructive Sleep Apnea

**DOI:** 10.3390/brainsci12101362

**Published:** 2022-10-08

**Authors:** Mi Lu, Wei Yu, Zhenjia Wang, Zhigang Huang

**Affiliations:** 1Department of Otolaryngology-Head and Neck Surgery, Beijing Tongren Hospital, Capital Medical University, Beijing 100005, China; 2Department of Radiology, Beijing Anzhen Hospital, Capital Medical University, Beijing 100029, China; 3Department of Radiology, Beijing Hospital of Traditional Chinese Medicine, Capital Medical University, Beijing 100010, China

**Keywords:** arousal, sleep fragmentation, coronary artery disease, obstructive sleep apnea, plaque, atherosclerotic, imaging

## Abstract

(1) Aim: We aim to evaluate the association between arousals during sleep and subclinical coronary atherosclerosis detected by coronary computed tomography angiography (CTA) in patients with obstructive sleep apnea (OSA). (2) Methods: This was a cross-sectional study. Consecutive newly diagnosed OSA patients, who underwent coronary CTA examinations within 3 months of the sleep study, were eligible. We used the arousal index (ArI) derived from polysomnography to assess arousals during sleep and a semi-automated plaque quantification software to characterize and quantify the subclinical coronary atherosclerosis. Multiple regression models were used to evaluate the associations of the ArI with the coronary atherosclerotic plaque presence, volume, and composition. (3) Results: A total of 99 patients with OSA were included in the study. In the multivariable models, patients with a high ArI (ArI > 32.2 events/h) were more likely to have coronary plaques compared to those with a low ArI (ArI ≤ 32.2 events/h) (OR: 3.29 [95% CI: 1.284 to 8.427], *p* = 0.013). Furthermore, the ArI exhibited significant associations with total (β = 0.015), noncalcified (β = 0.015), and low-attenuation (β = 0.012) coronary plaque volume after accounting for established risk factors (*p* = 0.008, 0.004, and 0.002, respectively). However, no association between the ArI and calcified plaque volume was found. (4) Conclusion: Repetitive arousals during sleep are associated with an increased coronary plaque burden in patients with OSA, which remained robust after adjusting for multiple established cardiovascular risk factors.

## 1. Introduction

Obstructive sleep apnea (OSA) affects nearly 1 billion individuals worldwide and is characterized by repetitive collapse of the upper airway during sleep, resulting in intermittent hypoxia and fragmented sleep [1]. Untreated OSA has been linked to coronary atherosclerotic plaque, myocardial infarction (MI), and metabolic disorders [2,3,4,5]. Although the underlying mechanisms are complicated and multifaceted, sympathetic activation is likely to be implicated in the increased risk of coronary events seen in OSA [6]. Recent studies demonstrated that sleep fragmentation, rather than recurrent hypoxia and reoxygenation, is the primary stimulus to sympathetic activation in OSA patients since sympathetic overactivity has a strong relationship with the arousal index (ArI) but not with the apnea–hypopnea index (AHI) or desaturation index [7,8,9]. Therefore, arousals during sleep may play a more important role in the initiation and progression of OSA-related coronary events than usually believed.

Studies exploring associations between arousals during sleep and subclinical atherosclerosis are scarce. Two existing studies found that the ArI was independently correlated with the presence of carotid plaque and a high coronary artery calcification burden, respectively [10,11]. No study has yet examined the effect of arousal frequency or other objective sleep metrics (e.g., slow-wave sleep percentage) on coronary plaque volume and composition. However, it is important to note that plaques causing MI or acute coronary syndromes (ACS) have specific compositional properties on coronary computed tomography angiography (CTA). Recent clinical studies found that low-attenuation plaque identified by coronary CTA had an excellent correlation with histologically confirmed inflamed plaque, which was the strongest predictor of subsequent MI [12,13]. Moreover, the noncalcified plaque burden predicted an MI better than the calcification score and stenosis [14]. All the above studies highlighted the relevance of plaque volume and composition in MI risk prediction.

Therefore, we aim to investigate whether repetitive arousals during sleep are associated with an increase in the presence, volume, and composition of coronary plaque in patients with OSA, independent of traditional cardiovascular risk factors, AHI, and nocturnal hypoxia.

## 2. Materials and Methods

### 2.1. Study Population

Consecutive patients who were newly diagnosed with OSA by polysomnography at Beijing Anzhen Hospital between September 2015 and October 2019 and who also underwent coronary CTAs within 3 months of the sleep studies were eligible for the current study. The patients were excluded if (1) they received OSA treatment during these 3 months; (2) they had other sleep disorders (e.g., insomnia, central sleep apnea); (3) they had a psychiatric condition (e.g., depression), and/or used antipsychotics in the last 3 months; (4) they had an MI history and/or underwent coronary revascularization; or (5) the original coronary CTA images were not available, or the quality of the image was poor.

Anthropometric measurements, medical history, and medication usage were obtained from a self-reported questionnaire and medical records. Body mass index (BMI) was calculated by dividing the body weight in kilograms by the square of height in meters. There were three categories of smoking status: never, former, and current smokers. The definitions of morbidities have been published in detail [2]. This study was approved by the Ethics Review Committee of Beijing Anzhen Hospital (No. 2017005), and written informed consent was obtained from all patients.

### 2.2. Sleep Study

Sleep recordings were performed using polysomnography (Grael, Compumedics, Abbotsford, VIC, Australia) at our sleep center. The sensors and recording montages mainly included electroencephalogram, bilateral electrooculograms, chin muscle electromyogram, electrocardiogram, oronasal thermistor, nasal pressure transducer/cannula, thoracic–abdominal inductance plethysmography, finger pulse oximetry, etc. Trained technicians attached the sensors and electrodes and scored the polysomnographic data according to the American Academy of Sleep Medicine scoring manual [15]. The average number of apneas and hypopneas per hour of sleep was used to determine the AHI and the patients with an AHI ≥ 5 events/h were diagnosed with OSA. Additionally, the percentage of sleep time spent in each sleep stage was also determined, namely non-rapid eye movement (NREM) stage 1 (N1%), NREM stage 2 (N2%), and NREM stage 3 (N3%, slow-wave sleep), and REM sleep (REM%). Arousal episodes were scored when there was an abrupt shift in the electroencephalogram frequencies (but not spindles) during sleep stages lasting at least 3 s, with at least 10 s of stable sleep before the change. The ArI was calculated as the number of arousals per hour of sleep. The sleep efficiency measure was the ratio between total sleep time and total bedtime.

### 2.3. Coronary CTA Image Acquisition

Coronary CTA was performed using prospectively electrocardiogram-triggered sequential scan techniques on a 256-row detector CT scanner (Revolution CT, GE Healthcare, Milwaukee, WI, USA) or a second-generation dual-source CT system (Somatom Definition Flash, Siemens Healthcare, Forchheim, Germany). A 50–60 mL dose of iodinated contrast agent (Ultravist, 370 mg iodine/mL; Bayer Schering Pharma, Guangzhou, China) was injected into an antecubital vein at a rate of 4–5 mL/s, followed by a 30–35 mL dose of saline solution. The scanning parameters were the gantry rotation time of 250–350 ms, a tube voltage of 100–120 kV depending on the BMI, and an automated tube current modulation. The bolus-tracking technique was used to trigger the acquisition of CT images, which started automatically after the CT value in the ascending aorta reached 120 HU. A median slice thickness of 0.625 mm and an increment of 0.5 mm were used to reconstruct the images. All procedures were performed following the Society of Cardiovascular Computed Tomography guidelines [16].

### 2.4. Coronary Plaque Analysis

An experienced radiologist (Z.W.) blinded to the clinical and sleep data, performed the quantitative measurement of coronary plaque using semiautomated software (Autoplaque Version 2.0, VoxelCloud Technology Co., Ltd., Suzhou, China). The overall method for plaque quantification of each coronary lesion was as follows. First, the proximal and distal limits of the plaque lesions were marked. Next, the plaque and vessel borders were automatically segmented, with manual adjustment performed if necessary. Finally, vessel volume and volume of calcified plaque, noncalcified plaque, and low-attenuation plaque (i.e., the noncalcified plaque with attenuation < 30 HU) were computed on a per-patient basis. Excellent inter-and intra-reader agreements for this software have previously been reported [17].

### 2.5. Statistical Analysis

Categorical variables are described as numbers with percentages and continuous variables as mean ± standard deviation (SD) or median (interquartile range) as appropriate. By dichotomizing the ArI values based on the cohort medians, we achieved low/high ArI groups (ArI ≤ 32.2 events/h and ArI > 32.2 events/h). Data between groups were compared using the independent-sample t-test, Mann–Whitney U test, χ^2^ test, or Fisher’s exact test, as appropriate. The Spearman’s rank correlation coefficients were obtained for the ArI with total, noncalcified, low-attenuation, and calcified coronary plaque volume, respectively. Multivariable stepwise logistic regression models were performed to test the independent associations of ArI and other sleep metrics—non-normally distributed variables—with the presence of coronary plaque in patients with OSA, by analyzing these objective sleep metrics as dichotomized variables (e.g., ArI > 32.2 events/h). The following covariates were adjusted: age, sex, BMI, smoking status, hypertension, diabetes, dyslipidemia, use of lipid-lowering medications, AHI, and lowest oxygen saturation (SpO_2_). Similarly, multivariable linear regression models were used to examine the independent association between ArI as the continuous variable and coronary plaque volume in patients with OSA. Since coronary plaque volumes exhibit a highly skewed distribution, they enter the linear models after logarithmic transformation. All p-values are two-sided with a significance threshold of 0.05. Statistical analyses were performed with SPSS software Version 19.0 (IBM Corp., Armonk, NY, USA). Figures were drawn using GraphPad Prism 9.

### 2.6. Sample Size Calculation

In this study, the coronary plaque volume was used as the outcome indicator, and multiple stepwise linear regression was performed to include 10 covariates. According to the principle of 10 outcome events per variable, the sample size should be around 100.

## 3. Results

### 3.1. Participants Characteristics 

A flow diagram illustrating study inclusion is shown in Figure 1. In total, 99 patients with OSA were finally included. The mean age was 50.6 (SD, 9.4) years; 76.8% were males. As shown in Table 1, patients with high ArI were more likely to be male, to have higher BMI, neck circumference, and diastolic blood pressure, to be sleepier, to have a higher AHI, oxygen desaturation index, percentage of nighttime with oxygen saturation < 90%, and a lower lowest SpO_2_ and mean SpO_2_ than those with low ArI.

A total of 52 patients (62.6%) had at least one coronary plaque, and the median total coronary plaque volume was 56.9 (interquartile range, 0–201.5) mm^3^. Participant characteristics stratified by the presence of coronary plaque are shown in Appendix A. In general, relative to individuals without coronary plaque, those with coronary plaques were more likely to be smokers, have a higher prevalence of hypertension, have greater usage of lipid-lowering medications, have a higher AHI and ODI, lower lowest SpO_2_ levels, more fragmented sleep as noted by the higher ArI, and a higher percentage of N1 and N2.

### 3.2. The Presence of Coronary Plaque

The prevalence of coronary plaque was higher in patients with a high ArI than in those with a low ArI, 37 (75.5%) versus 25 (50.0%), respectively (*p* = 0.009) (Table 1). Furthermore, we found that patients with an ArI > 32.2 events/h tended to have higher odds of having coronary plaque compared to those with a low ArI (crude OR: 3.083 [95% CI: 1.311 to 7.251], *p* = 0.01; adjusted OR: 3.29 [95% CI: 1.284 to 8.427], *p* = 0.013) (Table 2). However, there was no evidence for significant associations between adverse levels of sleep efficiency or percentage of time spent in slow-wave sleep and the presence of coronary plaque (all with *p*-value > 0.05).

### 3.3. The Volume and Composition of Coronary Plaque

Figure 2 shows the results of the volume and distributions of coronary plaque. We observed that the total plaque volume and volumes of noncalcified low-attenuation plaque were all significantly higher among patients with a high ArI than in those with a low ArI. Spearman’s correlation analysis showed that the ArI was significantly correlated with total (rho = 0.365, *p* < 0.001), noncalcified (rho = 0.386, *p* < 0.001), and low-attenuation (rho = 0.387, *p* < 0.001) coronary plaque volume, but it was not significantly correlated with calcified plaque volume (rho = 0.119, *p* = 0.241). Furthermore, linear regression models (Table 3) demonstrated that there were significantly positive associations between the ArI and total (β = 0.019), noncalcified (β = 0.019), and low-attenuation (β = 0.015) coronary plaque volume in the unadjusted models (all with *p*-value ≤ 0.001), and these associations remained after adjusting for multiple cardiovascular risk factors (β = 0.015, 0.015, and 0.012, respectively, and *p* = 0.008, 0.004, and 0.002, respectively). Conversely, no significant associations were observed between the ArI and the calcified plaque volume. In addition, the coronary plaque volume did not seem to be associated with sleep efficiency or the percentage of time spent in slow-wave sleep.

## 4. Discussion

This study was the first to evaluate the relationship between arousals during sleep and coronary plaque in patients with OSA. We found that not only was coronary plaque more likely to be present in patients with a high ArI (>32.2 events/h), but the total plaque volume and volumes of noncalcified and low-attenuation plaque increased with an increase in the ArI. Importantly, adjustment for potential confounders did not substantively alter these associations. In contrast, there was no evidence of a significant association between arousals and calcified plaque. These data suggest that the ArI, which quantifies the degree of fragmented sleep, may provide unique information about arousal-related coronary events risk in patients with OSA.

Cortical arousals are normal sleep phenomena; however, if they occur frequently, they can have acute and long-term effects on cardiovascular health. A recently published study found that frequent arousals were independently linked to hypertension in patients with OSA [18]. Meanwhile, in a community study with 8001 participants, Shahrbabaki et al. demonstrated that sleep arousal burden was associated with long-term all-cause and cardiovascular mortality, particularly in women [19]. However, we know little about the relationship between frequent arousals during sleep and atherosclerotic plaque in patients with OSA. In the current study, we not only highlighted the relationship between total coronary plaque volume and increased arousal frequency but also showed a high prevalence of coronary plaque in patients with a high ArI, which may provide evidence that atherosclerosis is an intermediate step in the pathophysiology from repetitive arousals to the development of adverse cardiovascular outcomes among OSA patients. To some extent, this finding was also similar to the results of a prior study using ultrasonography to measure the severity of carotid atherosclerosis in an observational study of 83 patients with OSA, which found that the ArI was one of the independent predictors of carotid artery plaque presence [10]. A similar pattern of association between high ArI levels and carotid plaque was also demonstrated by Zhao et al. in individuals aged <68 years [20]. However, the fact that the plaque components and their relationship to arousals were not evaluated in these two studies could be potential limitations because the alteration of the plaque composition would change plaque vulnerability. In general, the plaque with a large lipid core is more prone to rupture than those with calcified components (more stable) [21].

To further explore this issue, we investigated the association between the ArI and coronary plaque composition and observed that the ArI was an independent explanatory variable for noncalcified plaque volume and low-attenuation plaque volume. Total noncalcified plaque volume predicts ACS more accurately than clinical risk profiling or conventional coronary CTA readings (e.g., calcium score, luminal stenosis, and extent of coronary artery disease) alone, which has been well documented [22]. Previous studies also showed that most ACS cases are caused by the rupture and/or erosion of coronary plaques, which are typically characterized by a lipid-rich necrotic core covered by a thin fibrous cap [23,24]. Most importantly, these specific plaque components can be detected as low-attenuation plaque on coronary CTA [13]. Therefore, our findings can be interpreted as patients with frequent arousals having a high propensity to develop unstable plaque features, which may further be related to their increased risk of subsequent coronary events. It is, however, necessary to conduct further research to confirm this hypothesis.

In the current study, arousals during sleep were not significantly associated with the volume of calcified plaques. These results appear to be inconsistent with the findings from the Multi-Ethnic Study of Atherosclerosis, which found that among patients, with a mean age of 68 years, a high coronary artery calcium score (CAC > 400) was significantly correlated with a high ArI [11]. The potential explanation is that calcification is a well-recognized hallmark for the late stage of atherosclerosis. Specifically, plaque development involves several phases, including intimal inflammation, necrosis, fibrosis, calcification, arterial remodeling, fibrous cap rupture, and thrombosis, implying that the necrotic core calcifies over time [25]. However, the patients in our study were relatively young (average age was 50.6 years), and they may still be in the early stages of atherosclerosis.

The primary mechanism underlying the link between elevated arousal frequency and coronary plaque is possibly related to sympathetic nervous system activation and vascular endothelial dysfunction. Previous studies have shown that sympathetic activation led to increased concentrations of vasoconstrictor peptides and circulating catecholamines, which directly activate platelets [26]. The endpoint effect of persistent excessive platelet activation is myocardial infarction [27]. In addition, an animal study revealed that sleep fragmentation can contribute to atherosclerosis via altering hematopoiesis [28]. Mice subjected to sleep fragmentation produced less hypocretin (a wake-promoting neuropeptide) in the lateral hypothalamus, which increased the expression of colony-stimulating factor-1, stimulated monocyte differentiation into macrophages, and ultimately accelerated atherosclerosis.

We also evaluated whether the percentage of time spent in slow-wave sleep or sleep efficiency was associated with coronary plaque volume, but no statistically significant association was observed. These results agreed with the finding from another MESA study, where they also did not find an association between sleep efficiency or slow-wave sleep and carotid plaque in the overall cohort [20].

We acknowledge several limitations in our study. First, the sample size was small. Fortunately, the study was adequately powered to demonstrate the relationship between arousals (primary exposure) and subclinical coronary plaque burden, although not powered to detect associations between other sleep metrics (e.g., sleep efficiency) and coronary plaque. Second, due to the cross-sectional nature of our study, causality between the ArI and coronary plaque cannot be inferred. Third, in the present study, the patients with a high ArI had more potential confounding factors than those with low ArI, including BMI, AHI, etc. To minimize their confounding effects, we used multivariable stepwise regression models, which included these potential risk factors as covariates for statistical analysis. Fourth, there was a selection bias in this study. Considering that some of our patients were referred to sleep medicine centers for PSG studies after cardiology visits to control for predisposing factors for coronary heart disease, our cohort may not be fully representative of typical sleep clinic patients. Fifth, the objective sleep variables were derived from single-night polysomnography recordings. As a result, night-to-night variability in these sleep variables and the first night effect may exist and affect the strength of estimates. Sixth, we did not distinguish the types of arousal (e.g., respiratory, spontaneous, and movement-related arousal) and their association with coronary plaque, since the detailed arousal data were unavailable in our study. Finally, in addition to arousal frequency, arousal duration may also contribute to sleep fragmentation and be associated with cardiovascular mortality [19,29]. However, this metric was not measured in the present study.

## 5. Conclusions

In sum, frequent arousals during sleep were associated with an increased burden of coronary plaque in patients with OSA. These associations remained robust after considering multiple established cardiovascular risk factors, AHI, and nocturnal hypoxemia, which suggest that repetitive arousals from sleep might be novel and independent risk factors for subclinical coronary atherosclerosis. 

## Figures and Tables

**Figure 1 brainsci-12-01362-f001:**
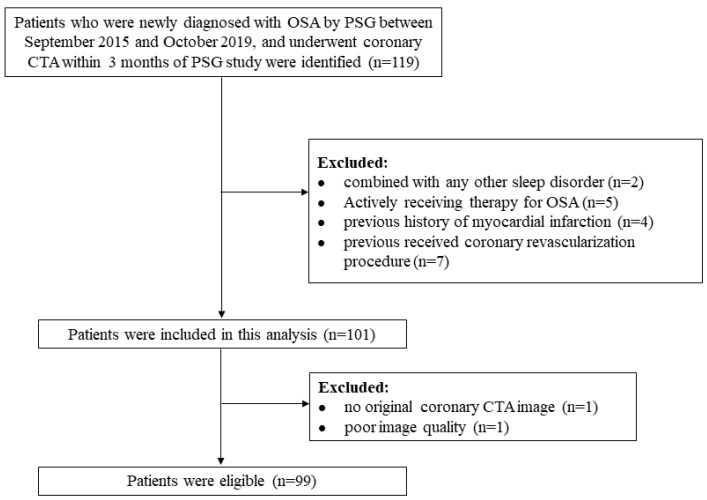
Flow diagram of the study. CTA = computed tomography angiography; OSA = obstructive sleep apnea; PSG = polysomnography.

**Figure 2 brainsci-12-01362-f002:**
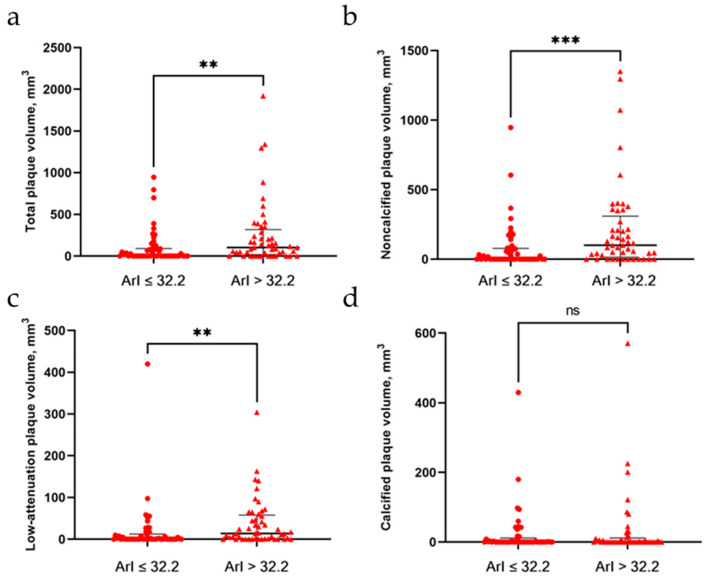
Volume and composition of coronary plaque between patients with a low ArI and patients with a high ArI. (**a**) Total plaque volume; (**b**) noncalcified plaque volume; (**c**) low-attenuation plaque volume; (**d**) calcified plaque volume. ** indicate *p*-Value <0.01; *** indicate *p*-Value <0.001. ArI = arousal index.

**Table 1 brainsci-12-01362-t001:** Baseline characteristics of the study population.

Variables	Overall (*n* = 99)	ArI ≤ 32.2 (*n* = 50)	ArI > 32.2 (*n* = 49)	*p*-Value
**Demographic and clinical characteristics**				
Age, years	50.6 ± 9.4	51.52 ± 10.1	49.7 ± 8.8	0.344
Male, *n* (%)	76 (76.8)	32 (64.0)	44 (89.8)	0.002
Body mass index, kg/m^2^	27.7 (25.8, 30.8)	26.4 (24.8, 27.8)	30.1(27.6, 32.3)	<0.001
Neck circumference, cm	40.9 ± 3.5	39.6 ± 3.3	42.4 ± 3.2	<0.001
Smoking status				0.541
Never, *n* (%)	52 (52.5)	29 (58.0)	23 (46.9)	
Former, *n* (%)	13 (13.1)	6 (12.0)	7 (14.3)	
Current, *n* (%)	34 (34.3)	15 (30.0)	19 (38.8)	
Systolic blood pressure, mmHg	133 (125, 147)	132(122.5, 145.5)	135 (128.5, 150)	0.173
Diastolic blood pressure, mmHg	83.2 ± 10.6	80.0 ± 9.9	86.4 ± 10.4	0.002
Hypertension, *n* (%)	56 (56.6)	28 (56.0)	28 (57.1)	0.909
Diabetes mellitus, *n* (%)	14 (14.1)	6 (12.0)	8 (16.3)	0.537
Hyperlipidemia, *n* (%)	35 (35.4)	19 (38.0)	16 (32.7)	0.578
Lipid-lowering medication use, *n* (%)	19 (19.2)	9 (18.0)	10 (20.4)	0.761
Prior history of CAD, *n* (%)	6 (6.1)	3 (6.0)	3 (6.1)	0.100
ESS	11 (8, 15)	9.0 (6.0, 12.5)	13 (9, 15)	0.006
**Polysomnography parameters**				
AHI, /h	36.5 (16.5, 61.4)	18.0 (10.8, 29.8)	60.0 (39.3, 72.3)	<0.001
ODI, /h	33.4 (16.5, 59.9)	17.5 (7.8, 28.8)	59.2 (41.5, 76.1)	<0.001
T90, %	3.5 (0.3, 13.1)	0.9 (0.1, 4.1)	10.8 (2.8, 24.1)	<0.001
Lowest SpO_2_, %	81.0 (72.0, 88.0)	86.0 (79.0, 89.0)	76.0 (66.0, 89.0)	<0.001
Mean SpO_2_, %	94.0 (91.0, 95.0)	95.0 (93.0, 96.0)	92.0 (89.0, 94.0)	<0.001
TST, min	409.0 (349.5, 447.5)	409.0 (347.6, 447.9)	409.0 (351.0, 442.8)	0.755
Sleep efficiency, %	78.0 (68.1, 84.4)	77.8 (68.1, 85.9)	78.4 (68.6, 83.7)	0.820
Time in sleep stage, % per TST				
N1	22.9 (16.2, 34.4)	16.3 (11.1, 20.9)	31.9 (26.2, 44.4)	<0.001
N2	48.0 (38.7, 52.6)	51.4 (44.1, 55.8)	42.2 (35.3, 48.7)	<0.001
N3	13.2 (5.5, 17.3)	15.6 (10.0, 21.4)	8.7 (3.9, 14.6)	<0.001
REM	15.6 (11.3, 20.3)	18.5 (13.1, 22.8)	12.9 (9.1, 17.3)	<0.001
**Coronary CTA measures**				
Presence of coronary plaque	62 (62.6)	25 (50.0)	37 (75.5)	0.009
Total plaque length, mm	13.4 (0, 28.1)	4.7 (0, 22.3)	20.8 (4.4, 35.8)	0.005
Total plaque volume, mm^3^	56.9 (0, 201.5)	9.2 (0, 90.8)	103.7 (18.1, 318.0)	0.002
Noncalcified plaque volume, mm^3^	47.8 (0, 173.3)	9.2 (0, 78.6)	100.8 (14.5, 309.5)	<0.001
Low-attenuation plaque volume, mm^3^	5.0 (0, 30.2)	0.3 (0, 12.4)	13.8 (0.6, 57.8)	0.001
Calcified plaque volume, mm^3^	0 (0, 10.4)	0 (0, 11.7)	0 (0, 11.8)	0.669

Data are presented as *n* (%), mean ± standard deviation, or median (interquartile range). AHI, apnea–hypopnea index; ArI, arousal index; CAD, coronary artery disease; ESS, Epworth sleepiness scale; ODI, oxygen desaturation index; REM, rapid eye movement; SpO_2_, oxygen saturation; TST, total sleep time; T90, percent of nighttime spent with oxygen saturation below 90%.

**Table 2 brainsci-12-01362-t002:** Logistic regression models assessing the association between sleep measures and the presence of coronary plaque.

	Unadjusted Odds Ratio (95% CI)	*p*-Value	Adjusted Odds Ratio * (95% CI)	*p*-Value #
Arousal index (>32.2 /h)	3.083 (1.311, 7.251)	0.010	3.290 (1.284, 8.427)	0.013
Sleep efficiency (>78.0%)	1.494 (0.658, 3.39)	0.337	NA	NA
Total sleep time (>409.0 min)	1.056 (0.467, 2.384)	0.896	NA	NA
Slow-wave sleep percentage (N3 > 13.2%)	1.056 (0.467, 2.384)	0.896	NA	NA

* Adjusted for age, sex, body mass index, smoking status, the presence of hypertension, diabetes, hyperlipidemia, lipid-lowering medications, AHI, and low SpO_2_. # Tested by multivariable backward stepwise logistic regression analyses. NA means the variable was not in the final model due to the corresponding *p*-Value > 0.1.

**Table 3 brainsci-12-01362-t003:** Linear models assessing the association between sleep measures and coronary plaque volume.

Dependent Variables	Independent variables	Unadjusted β ± SE	*p*-Value	Adjusted * β ± SE	*p*-Value #
Total plaque volume, mm^3^					
	Arousal index, /h	0.019 ± 0.06	0.001	0.015 ± 0.05	0.008
	Sleep efficiency, %	−0.001 ± 0.009	0.951	NA	NA
	Total sleep time, min	−0.001 ± 0.002	0.678	NA	NA
	Slow-wave sleep (N3), %	−0.007 ± 0.015	0.618	NA	NA
Non-calcified plaque volume, mm^3^					
	Arousal index, /h	0.019 ± 0.06	0.001	0.015 ± 0.05	0.004
	Sleep efficiency, %	−0.001 ± 0.009	0.912	NA	NA
	Total sleep time, min	−0.001 ± 0.002	0.608	NA	NA
	Slow-wave sleep (N3), %	−0.007 ± 0.014	0.624	NA	NA
Low-attenuation plaque volume, mm^3^					
	Arousal index, /h	0.015 ± 0.04	<0.001	0.012 ± 0.04	0.002
	Sleep efficiency, %	−0.002 ± 0.006	0.749	NA	NA
	Total sleep time, min	−0.001 ± 0.001	0.289	NA	NA
	Slow-wave sleep (N3), %	−0.007 ± 0.01	0.476	NA	NA
Calcified plaque volume, mm^3^					
	Arousal index, /h	0.003 ± 0.004	0.439	NA	NA
	Sleep efficiency, %	0.006 ± 0.006	0.306	NA	NA
	Total sleep time, min	0.001 ± 0.001	0.398	NA	NA
	Slow-wave sleep (N3), %	0.002 ± 0.01	0.867	NA	NA

* Adjusted for age, sex, body mass index, smoking status, the presence of hypertension, diabetes, hyperlipidemia, lipid-lowering medications, AHI, and low SpO_2_. # Tested by multivariable stepwise linear regression analyses. The coronary plaque volume data were entered into the models after logarithmic transformation as log (variable +1). NA means the variable was not in the final multivariable linear model due to the corresponding *p*-Value > 0.1.

## Data Availability

Not applicable.

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
