# Peer review of "Association between Arousals during Sleep and Subclinical Coronary Atherosclerosis in Patients with Obstructive Sleep Apnea"

_brainsci, 2022, doi:10.3390/brainsci12101362_

Round 1
Reviewer 1 Report
This manuscript is a cross-sectional study on the Association Between Arousals During Sleep and Subclinical Coronary Atherosclerosis in Patients with Obstructive Sleep Apnea.
It is an interesting and well-written paper.
I recommend its publication.
However, it needs minor improvements:
1.- Mat & Method
Sample size calculations are missing.
Lines 125-126: Why dichotomizing ArI values was the only statistical strategy? The authors should also use correlation and multiple regression, both with more power indeed.
2.- Discussion
Several confounding factors arise: age, neck circumference, BMI and diastolic BP were all significantly different between ArI<32 and ArI>32 (see table 1).
Also, AHI was very different: 18 vs 60!
Please, discuss in detail.
Limitations should include these possible confounding factors.
Reviewer 2 Report
The paper "Association Between Arousals During Sleep and Subclinical Coronary Atherosclerosis in Patients with Obstructive Sleep Apnea" by Mi Lu et al. is original, with good design, statistic methodology, results and discussions.
The main limitation is a no so large population (for a 4 years of recruitment), from a single center from one city in China.
Reviewer 3 Report
The study by Lu et al. provides interesting data on the association between OSA and atherosclerosis.
In the abstract ‘background’ should be changed to ‘aim’.
The background provides necessary information on ArI, which is important for further understanding of parameter choice for the analysis.
Yet, it would be advantageous to state that not only cardiovascular comorbidities are more frequent in OSA, but others as well, in particular, metabolic comorbidities (f.e. doi: 10.3390/jcm10173770).
Methods. Was usage of any other medications (f.e. benzodiazepines) considered as exclusion criteria? Were arousals grouped based on the reason – PLM, respiratory, other?
When stating who f.e. sored the study use initials instead of the full name.
Results. The data is well presented in tables and figures – clear for analysis.
Another mechanism of plaque generation should be mentioned/discussed (f.e. doi: 10.3389/fneur.2018.00635). Since hypoxia has a strong association discussion of other PSG parameters could be also included.
Reviewer 4 Report
Minor corrections to improve the overall quality:
- line 44, A surge of data has reproducibly identified strong associations of OSA with cardiac arrhythmias. Although most studies have focused on the links between OSA and atrial fibrillation (AF), relationships with ventricular arrhythmias have also been characterized. Key findings implicate OSA-related autonomic nervous system fluctuations typified by enhanced parasympathetic activation during respiratory events and sympathetic surges subsequent to respiratory events, which contribute to augmented arrhythmic propensity. Other more immediate pathophysiologic influences of OSA-enhancing arrhythmogenesis include intermittent hypoxia, intrathoracic pressure swings leading to atrial stretch, and hypercapnia, please discuss and cite doi:10.1016/j.chest.2016.09.014.
- line 46, medical CPAP or surgical therapy has demonstrated a protective role on cardiovascular events in OSA patients by reducing arrhythmic events, please discuss and cite doi:10.1016/j.amjoto.2021.103197
- line 46, medical cpap or surgical therapy have demonstrated a protective role on cardiovascular events in osa patients by reducing arrhythmic events. please discuss and cite doi:10.1016/j.chest.2016.09.014. - Which level of psg was performed? - add the consort flow diagram to specify the study protocol - the strobe guidelines where followed? - line 221, interesting data are reported in literature also for sleep fragmentation and arousals and pediatric osa. Please discuss and cite doi:10.3389/fnins.2021.644697 and doi:10.3390/children8100921
